# Recurrence of Equinus Foot in Cerebral Palsy following Its Correction—A Meta-Analysis

**DOI:** 10.3390/children9030339

**Published:** 2022-03-02

**Authors:** Axel Horsch, Matthias Claus Michael Klotz, Hadrian Platzer, Svenja Elisabeth Seide, Maher Ghandour

**Affiliations:** 1Department of Orthopedics and Trauma Surgery, Heidelberg University Hospital, 69120 Heidelberg, Germany; hadrian.platzer@med.uni-heidelberg.de (H.P.); mghandourmd@gmail.com (M.G.); 2Marienkrankenhaus Soest, Orthopedics and Trauma Surgery, 59494 Soest, Germany; mcmklotz@gmx.net; 3Institute of Medical Biometry, University of Heidelberg, 69117 Heidelberg, Germany; seide@imbi.uni-heidelberg.de

**Keywords:** recurrence, equinus, surgery, meta-analysis

## Abstract

Background: Recurrence in cerebral palsy (CP) patients who have undergone operative or non-operative correction varies greatly from one study to another. Therefore, we conducted this meta-analysis to determine the pooled rate of equinus recurrence following its correction either surgically or non-surgically. Methods: Nine electronic databases were searched from inception to 6 May 2021, and the search was updated on 13 August 2021. We included all studies that reported the recurrence rate of equinus following its correction among CP patients. The primary outcome was recurrence, where data were reported as a pooled event (PE) rate and its corresponding 95% confidence interval (CI). We used the Cochrane’s risk of bias (RoB-II) tool and ROBINS-I tool to assess the quality of included randomized and non-randomized trials, respectively. We conducted subgroup analyses to identify the sources of heterogeneity. Results: The overall rate of recurrence was 0.15 (95% CI: 0.05–0.18; *I*^2^ = 88%; *p* < 0.01). Subgroup analyses indicated that the laterality of CP, study design, and intervention type were significant contributors to heterogeneity. The recurrence rate of equinus differed among interventions; it was highest in the multilevel surgery group (PE = 0.27; 95% CI: 0.19–0.38) and lowest in the Ilizarov procedure group (PE = 0.10; 95% CI: 0.04–0.24). Twelve studies had a low risk of bias, eight had a moderate risk, and nine had a serious risk of bias. Conclusion: The recurrence of equinus following its correction, either surgically or non-surgically, in CP patients is notably high. However, due to the poor quality of available evidence, our findings should be interpreted with caution. Future studies are still warranted to determine the actual risk of equinus recurrence in CP.

## 1. Introduction

Cerebral palsy (CP) is the result of a non-progressive brain injury which occurs during the early stages of development [1]. Equinus deformity is one of the most frequent gait abnormalities that is seen in patients with CP. It is characterized by excessive ankle plantarflexion, especially during the stance phase [2]. Equinus deformity occurs mainly due to the contracture of the gastrocsoleus muscle; however, it can occur due to abnormalities in other muscles of the ankle joint, including the tibialis anterior and peroneus longus muscles [3,4,5]. There are two major types of equinus: dynamic and fixed equinus. Although dynamic equinus is mainly treated through nonoperative options, such as physiotherapy, Botulinum toxin A injection, serial casting, or ankle foot orthoses, fixed equinus is treated surgically by the lengthening of the gastrocsoleus muscle-tendon unit through a wide variety of approaches [6]. These approaches include single-event multilevel surgery (SEMLS), Tendo-Achilles Lengthening (TAL), and the Ilizarov procedure.

Recurrence following equinus correction is reported in numerous studies with rates differing widely from one study to another (ranging from 4.7 to 28.2%). There is a wide discrepancy in the reported rates of recurrence, with no consistency in the rates reported in individual studies. In the same context, according to a recent systematic review, the recurrence rate of equinus deformity following TAL surgery was reported to range from 0 to 43% [7], with similar rates among unilateral and bilateral spastic CP patients who had undergone gastrocsoleus lengthening (0–38% and 0–35%, respectively) [8]. It is noteworthy that the rate of recurrence differs substantially among studies, and this warrants further investigation to determine the actual rate of recurrence of equinus following its correction. We hypothesize that the recurrence rate of equinus following its correction will be notably high, both in surgically and non-surgically treated CP patients.

Thus, we carried out this research to estimate the rate of equinus recurrence following its correction, either surgically or non-surgically, among CP patients. We also aimed to identify the current gaps in the literature regarding equinus recurrence following its correction.

## 2. Materials and Methods

### 2.1. Search Strategy and Study Selection

This study was carried out in accordance with the PRISMA guidelines for systematic reviews and meta-analyses, where registering a protocol is not considered mandatory [9]. We performed a database search through several databases: PubMed, Scopus, ScienceDirect, Clinicaltrials.gov (20 January 2022), CENTRAL, Virtual Health Library (VHL), Global Health Library (GHL), NYAM, and SIGLE. The search was originally carried out on 6 May 2021, and it was later updated on 13 August 2021, yielding no more relevant papers.

The following keywords were used when performing the search: (recurrence) AND (equinus) AND (“cerebral palsy”). Medical subject headings (MeSH) were also used to include a wider pool of studies. In an attempt to ensure that all relevant studies were included, we carried out a manual search of references through several approaches: searching for similar articles on PubMed, going through the reference list of included studies, and searching Google manually [10,11] Appendix A. The search process was conducted in accordance with the PICO framework: participants were patients with equinus who underwent operative or non-operative correction, the intervention was any type of management approach used for the correction of equinus, the comparison was any type of surgery or other treatment approaches (i.e., casting, injection), and recurrence was the outcome of interest.

We included all original studies that reported the recurrence of equinus following its correction (surgically or non-surgically), whether there was a comparison group or not. We excluded papers if they had one of the following criteria: (1) non-original research, (2) in vitro and animal research, (3) duplicates, (4) studies with overlapping datasets, (5) unextractable data due to selective reporting or unclear data, (6) studies with unavailable full texts, and (7) secondary research (i.e., reviews, editorials, commentaries, correspondence, etc.). Four authors carried out the title/abstract screening phase. Then, the full text of relevant articles was retrieved, and three authors screened the retrieved manuscripts. Any disagreements between authors were solved through thorough discussion, and if a solution was not reached a senior author was consulted. The PRISMA flow diagram of the study process is illustrated in Figure 1.

### 2.2. Data Extraction

A data extraction sheet was made using the Excel software. First, a pilot extraction was conducted to include all of the necessary components in the extraction sheet. In summary, the sheet included three parts, where each part was focused on a certain aspect of the included studies. The first part contained the baseline characteristics of included studies, such as the title of the study, authors’ names, year of publication, journal, etc. The second part contained the demographic characteristics of patients included in the individual studies, such as sample size, age and gender, study design, type of equinus or CP, type of intervention performed, etc. The third part focused on the outcome (i.e., rates of recurrence following correction). Two authors conducted the data extraction process, while a third author checked all of the extracted data to ensure their accuracy. Any disagreements between authors were solved by discussions and consensus.

### 2.3. Risk of Bias

We checked the methodological quality of the included studies through the use of Cochrane’s revised risk of bias tool [12] and the risk of bias in non-randomized studies of intervention (ROB 2) tool [13] for randomized and non-randomized trials, respectively. This process was carried out by three authors, and any discrepancy was solved by discussion amongst them.

### 2.4. Statistical Analysis

Overall pooled proportions were estimated using a generalized linear mixed model and a logit transformation to pool the recurrence rate [14]. The overall recurrence of equinus was reported in the form of the pooled estimate (PE) and its corresponding 95% confidence interval (CI). Meta-analyses were conducted using the random-effects model, where the DerSimonian–Laird estimator was used to report the between-trial heterogeneity [15]. In this study, between-trial heterogeneity is reported by using the estimated variance $\tau^{2}$ and the $I^{2}$, and the meta-analysis results are visualized through forest plots. We further conducted several subgroup analyses based on a set of variables: laterality of CP, study design, definition of equinus, and type of intervention to correct equinus. Analyses were performed using R [16] version 4.0.3 and its extension, meta [17].

## 3. Results

### 3.1. Search Results

The initial database search yielded 508 articles, of which 410 were included in the title/abstract screening phase following the exclusion of duplicated articles through the EndNote software. Fifty-seven articles were eligible for full-text screening, of which twenty-eight articles were excluded for the following reasons: reviews (N = 21), unavailable full texts (N = 3), and abstract-only papers (N = 4). Ultimately, a total of 29 articles were included in our review and analysis.

### 3.2. Study Characteristics

The baseline characteristics of included studies are presented in Table 1. 

In terms of study design, 18 articles were retrospective cohort studies [6,8,12,13,16,17,18,19,20,22,23,25,26,32,33,40,41,46], 6 were case series [18,19,20,21,22,23], 2 were cohort studies (unidentifiable as prospective or retrospective) [24,25], 2 were prospective cohort studies [26,27], and 1 was a randomized controlled trial [28]. The overall sample size was 2071 patients, ranging from 9 cases [23] to 347 cases [29]. Gender was reported in 12 studies, and males were predominant in all studies. The follow-up of patients varied among included studies, ranging from 6 months [28] to 168 months [30].

Individual studies reported different interventional approaches for the correction of equinus deformity, including Tendo-Achilles lengthening (TAL, both percutaneous and open), multilevel surgery, tibialis anterior tendon shortening, the Vulpius procedure, the aponeurectomy of gastrocnemius muscle, gastrocnemius recession, Ilizarov external fixation, Botox injections, casting, Baumann’s method, and Baker’s method. However, only six studies had a control or a comparison group [27,28,30,38,39,41].

The laterality of cerebral palsy was reported in 24 studies: bilateral in two studies [24,31] and both unilateral and bilateral in twenty-two studies [1,2,4,5,6,8,13,16,17,18,19,20,22,23,25,26,30,32,33,35,41,42]. The type of cerebral palsy was reported in thirteen studies, where one study reported a mixed CP type [41]. Twelve studies reported spastic CP [18,21,24,25,28,30,31,32,35,43,44,45], and one study reported athetoid CP [25]. The type of equinus was reported six studies, where one study reported static equinus [21] and five studies reported fixed equinus [18,26,31,32,33].

### 3.3. Risk of Bias

Based on the quality assessment of the single included RCT (based on five domains), the study was deemed to have a low risk of bias [28]. Based on the assessment of non-randomized studies of intervention (based on seven domains), eleven studies had a low risk of bias [18,21,22,23,25,26,27,35,36,38,42], eight studies had a moderate risk of bias [19,29,30,33,34,39,43,46], and nine studies had serious risk of bias [8,11,12,13,16,20,23,25,35]. The most common domain that had a serious risk of bias was confounding.

### 3.4. Overall Recurrence of Equinus

A total of 29 studies were included in the meta-analysis, and the overall rate of the recurrence of equinus following interventional correction of any type was 0.15 (95% CI: 0.10–0.20) (Figure 2). However, there was significant heterogeneity (*I*^2^ = 88%; *p* < 0.01). Six studies included a comparator group (Vulpius procedure, open TAL, serial casting, and Hoke’s method), and, upon comparison of the intervention and comparator groups no significant difference in the rate of recurrence was noted (OR = 0.85; 95% CI: 0.36–2.06). Additionally, significant heterogeneity was encountered (*I*^2^ = 73%; *p* < 0.01) (Appendix A). Due to the absence of a standard control group, no further discussion about the comparative risk/odds of equinus recurrence will be attempted.

### 3.5. Subgroup Analysis

In an attempt to determine the causes of heterogeneity in our analysis, various subgroup analyses were conducted based on laterality, study design, equinus definition, and intervention type.

### 3.6. Recurrence and Laterality

Laterality was a significant contributor to heterogeneity, where studies that reported both unilateral and bilateral disease were significantly heterogenous (*I*^2^ = 85%; *p* < 0.01), while studies reporting bilateral disease resulted in no heterogeneity (*I*^2^ = 0%; *p* = 0.22). That being said, the recurrence rate was different in studies reporting both laterality and bilaterality (pooled estimated (PE) = 0.16; 95% CI: 0.12–0.22) to those reporting bilateral disease alone (PE = 0.28; 95% CI: 0.21–0.36) (Figure 3).

### 3.7. Recurrence and Study Design

Study design was also a contributor to heterogeneity where case series did not result in significant heterogeneity (*I*^2^ = 38%; *p* = 0.38). In addition, the recurrence rate of equinus did not differ much between the different groups, where case series and retrospective cohort studies had an overall rate of 0.10 (95% CI: 0.05–0.18) and 0.17 (95% CI: 0.11–0.26) (Figure 4).

### 3.8. Recurrence and Equinus Definition

Equinus was defined in six studies [18,21,26,31,32,33], and even though the rate of recurrence was not different among studies that reported a definition of equinus (PE = 0.18; 95% CI: 0.09–0.34) and those that did not (PE = 0.14; 95% CI: 0.09–0.20), the heterogeneity remained significant (*I*^2^ = 82%; *p* <0.01, *I*^2^ = 87%; *p* < 0.01, respectively) (Figure 5).

### 3.9. Recurrence and Intervention Type

Different interventions were reported in included studies, such as the Ilizarov procedure (N = 2 studies) [18,26], Tendo-Achilles Lengthening (N = 12 studies) [19,20,23,27,29,32,34,37,39,40,41,45], multilevel surgery (N = 4 studies) [24,25,33,38], and casting (N = 2 studies) [28,44].

The subgroup analysis based on the type of intervention (surgery) yielded no resultant heterogeneity in the Ilizarov method and casting subgroups (*I*^2^ = 0%; *p* = 0.24 and *I*^2^ = 0%; *p* = 0.44, respectively). That being said, the rate of the recurrence of equinus differed in different subgroups, where it was 0.10 (95% CI: 0.04–0.24), 0.17 (95% CI: 0.11–0.25), 0.27 (95% CI: 0.19–0.38), and 0.17 (95% CI: 0.09–0.31) in the Ilizarov, TAL, multilevel surgery, and casting subgroups, respectively (Figure 6).

## 4. Discussion

This is the first systematic review and meta-analysis to examine the rate of equinus recurrence following its correction by different types of surgeries. This study is considered a hypothesis-generating review aimed at identifying the gaps in the current evidence in order to better determine the actual risk of the recurrence of equinus following its management. Four different interventional approaches are reported in this review: the Ilizarov procedure, Tendo-Achilles Lengthening (TAL), multilevel surgery, and casting. Multilevel surgery was defined as undergoing more than one procedure at the same time. For example, in the study of Matsuo et al. [25] patients underwent the following procedures: gastrocnemius aponeurotic lengthening, intramuscular lengthening of the peroneus longus and tabialis posterior, and sliding lengthening of the flexor hallucis longus and flexor digitorum longus.

Overall, the pooled recurrence rate in our meta-analysis was 15%, which differed substantially based on laterality, study design, equinus definition (yes/no), and the type of intervention. For instance, the pooled rate of recurrence of equinus was higher in bilateral disease (28%) compared to both unilateral and bilateral disease (16%). Case series had lower rates of recurrence compared to retrospective cohort design (10% vs. 17%). The difference between studies that reported a clear definition of equinus and those that did not, in terms of recurrence rates, was minimal (18% vs. 14%, respectively). It is of note that casting and TAL had the same recurrence rate of 17%, while the Ilizarov procedure resulted in a lower rate (10%) and multilevel surgery was associated with the highest rate of recurrence (27%).

That being said, we encountered significant substantial heterogeneity in our analyses. Therefore, we conducted several subgroup analyses to determine the causes of the resultant heterogeneity. We carried out a subgroup analysis based on laterality, study design, equinus definition (defined/not), and intervention type. It is worth noting that laterality, study design, and intervention types were significant contributors to heterogeneity where bilateral disease, case series, the Ilizarov method, and casting procedures had no significant heterogeneity (*I*^2^ < 40%).

It is noteworthy that the follow-up duration of equinus patients varied remarkedly between individual studies, ranging from 6 months to as long as 168 months after the interventional correction of equinus. Moreover, based on the qualitative assessment of retrieved data from individual studies, we noted a tendency towards higher recurrence rates in studies with longer follow-up durations. This further complicates the issue of the actual recurrence rate after equinus correction in CP patients. Therefore, we recommend future studies report the recurrence rate in both the short and long term.

Our findings are quite different from what has been reported in the literature. In 2011, Shore et al. [7] conducted a systematic review aimed at determining the different operative options for the correction of equinus in patients with cerebral palsy. A total of 23 studies reported the use of TAL for the correction of equinus deformity, of which 14 studies reported the recurrence rate of equinus after surgery, which ranged from 0% to 43%. Similarly, in the recently published systematic review of Ma et al. [8], the authors investigated the recurrence rates of equinus in patients with cerebral palsy who underwent gastrocsoleus lengthening. The authors reported recurrence rates of 0–38% and 0–35% among patients with unilateral spastic and bilateral spastic cerebral palsy, respectively. In our study, the recurrence rate of TAL was 17%. This difference could be related to numerous factors. For instance, studies that had longer follow-up times reported much higher recurrence rates [33,45]. Other factors include age at the index date (time of undergoing surgery), the typographical distribution of CP, and the use of postoperative casting [47].

## 5. Limitations

Our study is the first to examine the rate of equinus recurrence in different interventions for the correction of equinus in patients with cerebral palsy. We highlight that many factors could play a role in the rate/risk of equinus recurrence following its correction, including clinicodemographic characteristics, such as laterality and the type of intervention used. However, our review encountered several limitations. First, there is a clear lack of randomized controlled trials that investigate the recurrence rate of equinus following either intervention (surgical or non-surgical), and this may be, in part, due to the long follow up period that is required to diagnose recurrence. That being said, available cohort studies are of poor quality, with 17 studies having a moderate to serious risk of bias. Notably, the investigation of potential confounders is lacking in most studies. Therefore, future work should take into consideration the potential confounding variables that might affect the recurrence rate. Second, there is no standardized definition or diagnostic criteria for equinus recurrence, which could potentially contribute to the wide variability in recurrence rates among studies. Therefore, scholars are advised to clearly state the criteria they used for defining recurrence. Third, due to the absence of a standardized control group, we could not analyze the comparative risk of developing recurrence among different treatment groups. This needs to be addressed in future randomized controlled trials. Fourth, the resultant heterogeneity in our analysis limits the validity and generalizability of our findings; therefore, more work is needed to investigate the potential patient- or intervention-related sources of heterogeneity. Finally, due to the lack of relevant data, we could not analyze the predictors of the recurrence of equinus. Notably, only four studies in our review classified patients with CP according to their GMFCS levels [21,24,29,45], despite the value of GMFCS being the best approach in deciding for the correction of equinus in CP patients. We hypothesize that outcomes, particularly recurrence, could vary considerably across different GMFCS levels; therefore, future studies are warranted to investigate this matter.

## 6. Conclusions

The rate of recurrence of equinus following surgery in CP patients is notably high. However, due to the poor quality of available evidence and the presence of multiple limitations, our findings should be interpreted with caution. Our review can be used for hypothesis generation, and more studies are still needed to determine the actual rate of equinus recurrence in CP.

## Figures and Tables

**Figure 1 children-09-00339-f001:**
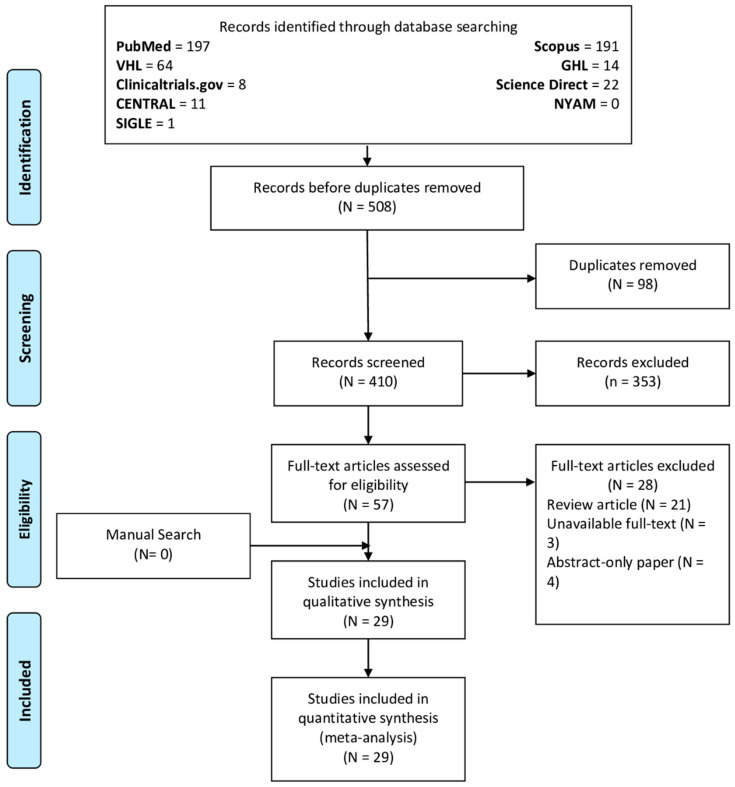
PRISMA flow diagram of the review process.

**Figure 2 children-09-00339-f002:**
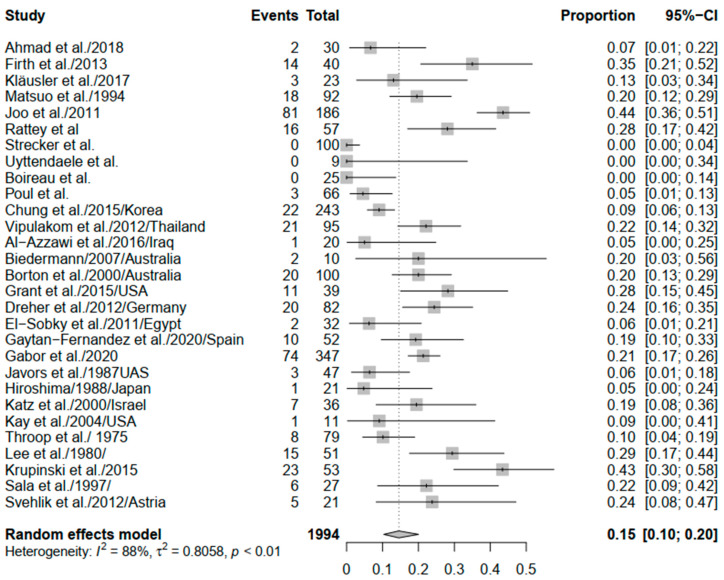
Forest plot of the overall recurrence of equinus.

**Figure 3 children-09-00339-f003:**
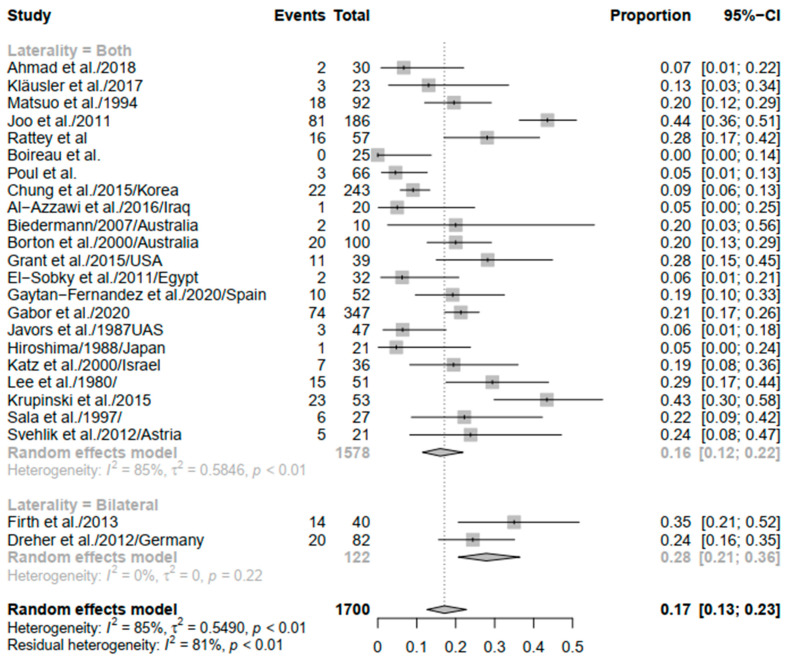
Forest plot of subgroup analysis of equinus recurrence based on the laterality of CP.

**Figure 4 children-09-00339-f004:**
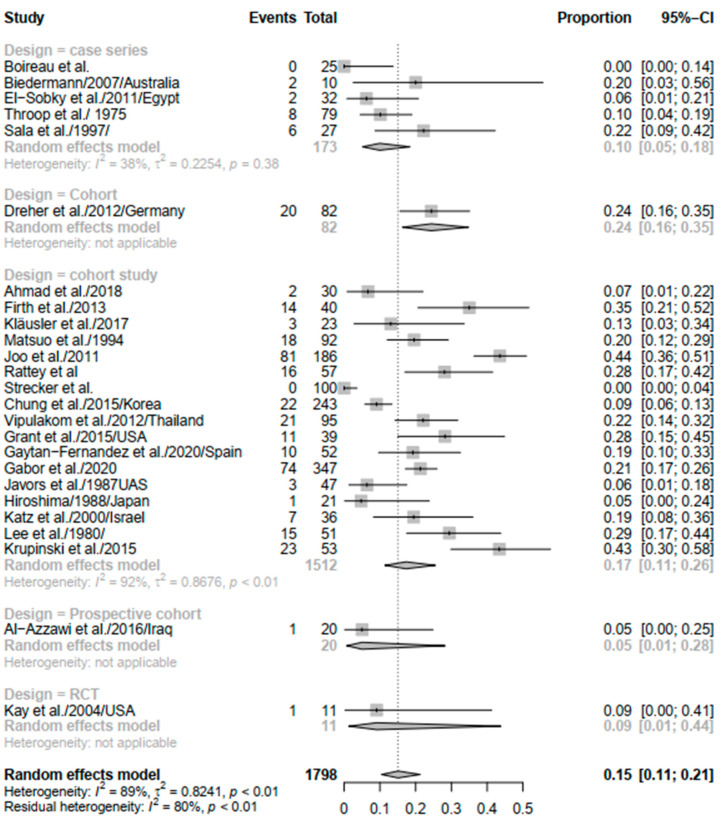
Forest plot of subgroup analysis of equinus recurrence based on study design.

**Figure 5 children-09-00339-f005:**
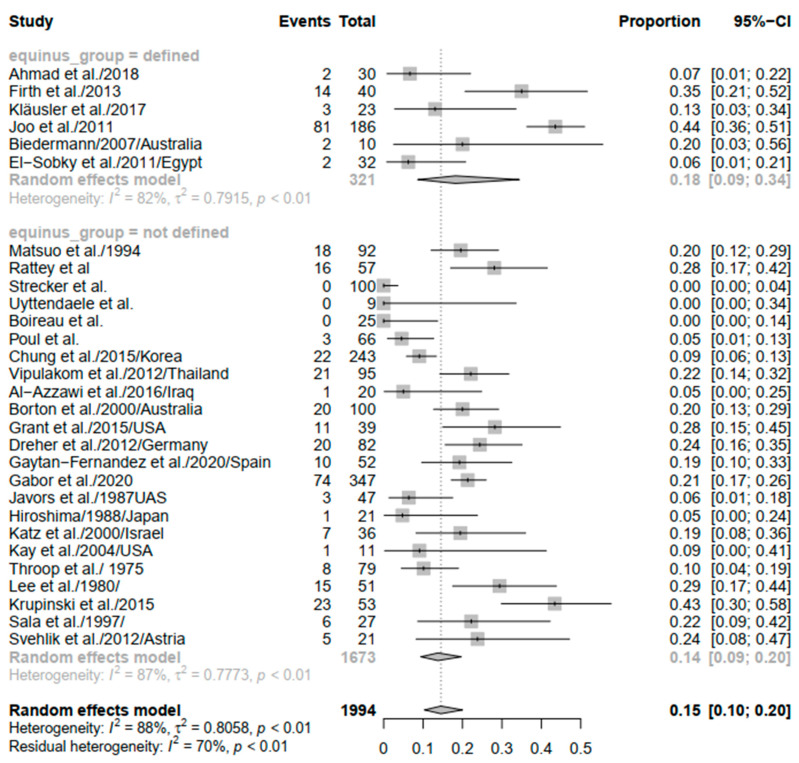
Forest plot of the subgroup analysis of equinus recurrence based on equinus definition.

**Figure 6 children-09-00339-f006:**
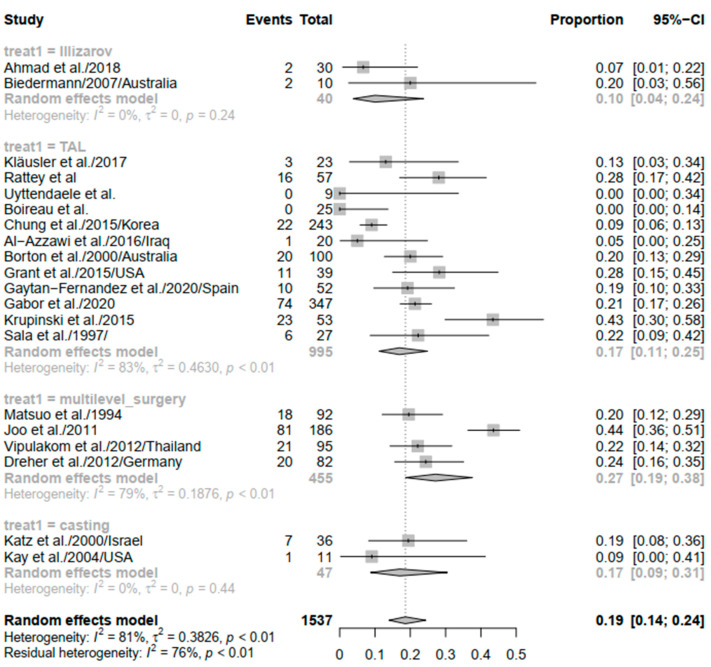
Forest plot of the subgroup analysis of equinus recurrence based on intervention type.

**Table 1 children-09-00339-t001:** Baseline characteristics of studies included in the systematic review (N = 29).

Author/YOP	Design	Size	Laterality	CP Type	Equinus Type	Intervention	Control	Age: Mean (SD)	Male/Female	Follow up (Months)
Biedermann/2007 [18]	Case series	10	Both	Spastic	Fixed	Ilizarov external fixation	None	12.3	8/2	50
Boireau/2002 [19]	Case series	25	Both	NR	NR	Percutaneous lengthening of the Achilles tendon	None	NR	NR	NR
Sala/1997 [20]	Case series	27	Both	NR	NR	TAL	None	NR	15/12	66
El-Sobky/2011 [21]	Case series	21	Both	Spastic	Static	Open distal gastrocnemius recession	None	5–14 *	13/8	26
Throop/1975 [22]	Case series	48	NR	NR	NR	Murphy procedure	None	NR	NR	36
Uyttendaele/1989 [23]	Case series	9	NR	NR	NR	Combined Achilles and tibialis posterior lengthening	None	NR	NR	NR
Dreher/2012 [24]	Cohort	44	Bilateral	Spastic	NR	Multi-level surgery	None	9.8 (3.4)	NR	108
Matsuo/1994 [25]	Cohort study	92	Both	spastic/athetoid	Surgery	None	3–19 *	NR	50	
Ahmad/2018 [26]	Prospective cohort	30	Both	NR	Fixed	Ilizarov external fixation	None	3–18 *	21/9	18
Al-Azzawi/2016 [27]	Prospective cohort	22	Both	NR	NR	Percutaneous TAL	Open TAL	4–10 *	14/8	12
Kay/2004 [28]	RCT	23	NR	Spastic	NR	Botox plus casting	serial casting alone	4.3–13.8 *	12/11	6
Gabor/2020 [29]	Retrospective Cohort	347	Both	NR	NR	Percutaneous (261 cases) and open TAL (86 cases)	None	2.8–15.1 *	NR	NR
Lee/1980 [30]	Retrospective Cohort	116	Both	Spastic	NR	Baker’s method (tongue-in-groove incision in the gastrocnemius aponeurosis)	Hoke’s method (tendo-calcaneus lengthening)	NR	NR	168
Firth/2013 [31]	Retrospective cohort	40	Bilateral	spastic	Fixed	Gastrocnemius lengthening	None	10	25/15	90
Kläusler/2017 [32]	Retrospective cohort	23	Both	spastic	Fixed	Tibialis Anterior Tendon Shortening plus TAL	None	14.9	13/8	70
Joo/2011 [33]	Retrospective Cohort	186	Both	NR	Fixed	Surgery	None	6.8	118/68	135.5
Rattey/1993 [34]	Retrospective Cohort	57	Both	NR	NR	TAL by open Z-plasty	None	NR	NR	120
Strecker/1990 [35]	Retrospective Cohort	100	NR	Spastic	NR	Anterior transposition of the Achilles tendon	None	NR	NR	30
Poul/2003 [36]	Retrospective Cohort	61	Both	NR	NR	Percutaneous aponeurectomy of the gastrocnemius	None	NR	NR	36
Chung/2015 [37]	Retrospective Cohort	243	Both	NR	NR	TAL	None	7.8 (2.7)	159/84	97
Vipulakom/2012 [38]	Retrospective Cohort	95	NR	NR	NR	TAL	Vulpius procedure	2–19.9 *	NR	NR
Borton/2000 [39]	Retrospective cohort	195	Both	NR	NR	Percutaneous TAL	Open TAL and Baker’s method	2–18 *	113/82	NR
Grant/2015 [40]	Retrospective Cohort	27	Both	NR	NR	TAL	None	2–9 *	NR	120
Gaytan-Fernandez/2020 [41]	Retrospective Cohort	55	Both	Mixed	NR	Percutaneous TAL	Open TAL	1–16 *	45/19	72
Javors/1987 [42]	Retrospective Cohort	47	Both	NR	NR	Vulpius procedure	None	2–14 *	NR	68.4
Hiroshima/1988 [43]	Retrospective Cohort	21	Both	Spastic	NR	Surgery (Anterior transfer of long toe flexors)	None	NR	NR	NR
Katz/2000 [44]	Retrospective Cohort	36	Both	Spastic	NR	Below-knee cast plus early weight bearing without splint or orthosis	None	4–6 *	NR	75
Krupinski et al./2015 [45]	Retrospective Cohort	53	Both	Spastic	NR	Subcutaneous TAL	None	7	NR	121
Svehlik/2012 [46]	Retrospective Cohort	18	Both	NR	NR	Baumann’s method (fractional lengthening of the gastrocnemii and soleus muscles)	None	NR	NR	120

* data provided in these studies refer to the range of age of participants included; TAL: Tendo-Achilles Lengthening; NR: Not Reported; SD: Standard Deviation; CP: Cerebral Palsy; YOP: Year of Publication.

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
