# Peer review of "Recurrence of Equinus Foot in Cerebral Palsy following Its Correction—A Meta-Analysis"

_children, 2022, doi:10.3390/children9030339_

Round 1
Reviewer 1 Report
To the authors,
The recurrence rate after surgery is a very interesting topic for doctors or surgeons who care and operate on the equinus deformity of cerebral palsy patients.
I think the idea of this study to compare the recurrence rate of equinus foot surgery is novel.
However, the comparison of the results of the papers will not be consistent without the definition of equinus status in patients with cerebral palsy, the limitation of the ankle range of motion before surgery, the degree of spasticity before and after surgery, the degree of range of motion after surgery.
In addition, the ankle joint contracture with spasticity is affected by the growth rate in the pediatric patient. As the authors described in the text, it can also be affected by CP type, diplegia, or hemiplegia.
Therefore, the study itself could not produce clear results about the recurrence rate, but the results of this study suggest that doctors considering surgical treatment for cerebral palsy in the future, need consensus on checkpoints that need to be identified or considered in advance.
As reviewers of this paper, I hope that people who read this research paper in the future will be reminded of this point.
As a minor revision, it would be better to change the font size of the letters or change the width of the column to make it more readable for the horizontal alignment of words in Table 1.
Thank you.
Author Response
Comment 1: To the authors, the recurrence rate after surgery is a very interesting topic for doctors or surgeons who care and operate on the equinus deformity of cerebral palsy patients. I think the idea of this study to compare the recurrence rate of equinus foot surgery is novel. However, the comparison of the results of the papers will not be consistent without the definition of equinus status in patients with cerebral palsy, the limitation of the ankle range of motion before surgery, the degree of spasticity before and after surgery, the degree of range of motion after surgery.
Response: Thank you for your helpful insights in making this work of better quality. Regarding the first point, there is a clear lack in a standardized definition criterion for equinus, which further complicates the interpretation of the findings in the literature regarding equinus prevalence, improvement after correction, or recurrence. Even though, in our analysis, we divided the data based on (defined equinus vs. non-defined equinus), there is a clear heterogeneity (I2 = 83%), and this is a result of the inconsistent definition criteria for equinus in included studies. Therefore, in our previous version of the manuscript, we highlighted this issue in the limitation section, and now, we are currently in the process of publishing an original research manuscript tackling the problem of inconsistent equinus definition/diagnosis in order to reach a near-valid and reliable cut-off value of dorsiflexion (when the knee is extended) for defining equinus foot.
As for the 2nd comment, unfortunately, data regarding pre- vs. post-operative range of motion or spasticity before/after intervention was lacking, and that is why we couldn’t perform an analysis on these variables.
Comment 2: In addition, the ankle joint contracture with spasticity is affected by the growth rate in the pediatric patient. As the authors described in the text, it can also be affected by CP type, diplegia, or hemiplegia. Therefore, the study itself could not produce clear results about the recurrence rate, but the results of this study suggest that doctors considering surgical treatment for cerebral palsy in the future, need consensus on checkpoints that need to be identified or considered in advance. As reviewers of this paper, I hope that people who read this research paper in the future will be reminded of this point.
Response: Thank you for your comment. We totally agree with your suggestion given the great discrepancy in this disease entity, regarding its diagnosis, management, and follow-up. We added a new sentence to further stress on the implications of our study: [Our study is the first to examine the rate of equinus recurrence in different interventions for the correction of equinus in patients with cerebral palsy. We highlight that many factors could play a role in the rate/risk of equinus recurrence following its correction, including clinicodemographic characteristics, such as laterality and the type of intervention.]
Comment 3: As a minor revision, it would be better to change the font size of the letters or change the width of the column to make it more readable for the horizontal alignment of words in Table 1.
Response: Thank you for your help. Initially, the Table was in a “landscape view”, and now we have modified it to fit the “portrait view”. We also added a description for all of the abbreviations provided in the Table.
Reviewer 2 Report
The aim of this paper is to systematic review and meta-analyze the rate of equinus recurrence following its operative correction among patients with CP.
This is a relevant topic for clinicians and orthopedics treating children with Cerebral Palsy. As equinus foot is a common condition in CP, and surgery is mostly performed, it is very important to provide evidences about the effectiveness of surgical treatment.
However, there are some critical issues that make the manuscript in its present form not acceptable.
Major issues:
MATERIALS and METHODS line 66-67: authors declare that the articles included only articles about “operative corrections” and “any type of surgery”. We noticed that you included also the article of Kay et al ( table 1) where the patients didn’t receive any kind of surgery but only serial casting or casting+ Botox: we can’t consider these procedures as surgical, so we think that this article should be excluded by the metanalysis, whose objective is the evaluation of the recurrence of equinus
Discussion, page 9: please discuss the wide range of follow up, recurrence can be related to the assessment time after surgery.
Furthermore we suggest to proceed on the possible analysis of subgroups: as recurrence is more common during growth, why didn’t you consider to divide groups according age and follow-up lenght?
Minor points
- Line 34-35: “ Equinus deformity does not occur particularly due to contracture of the gastrocsoleus muscle” is it an error or do you mean that there is another more common cause of it?
- Table 1 pag 5: you should describe the Intervention of Matsuo 2011 and Joo 2011
- Table 1 pag 6: as multilevel surgery is not specific for equinus deformity, you should specify which is the specific intervention considered by Dreher 2012 for the equinus. The same is for Katz 2000.
Author Response
Major issues
Comment 1: MATERIALS and METHODS line 66-67: authors declare that the articles included only articles about “operative corrections” and “any type of surgery”. We noticed that you included also the article of Kay et al ( table 1) where the patients didn’t receive any kind of surgery but only serial casting or casting+ Botox: we can’t consider these procedures as surgical, so we think that this article should be excluded by the metanalysis, whose objective is the evaluation of the recurrence of equinus
Response: Thank you for highlighting this mistake. Originally, this study was conducted to discuss the recurrence rate of equinus in all interventions used to correct this complication of CP, and this is evident from our search criteria (recurrence) AND (equinus) AND (“cerebral palsy”), where we didn’t include a limitation based on the type of intervention. Because the majority of studies were reporting surgery over other non-surgical interventions, we focused the main theme of our review on surgery.
Now, in the revised version of the manuscript we have edited the manuscript, including the Title, to fit the “correction of equinus” in general rather than focusing on “surgical correction” alone. Also, in the results and discussion sections, the presentation of data based on the type of intervention further clarifies and supports our approach. All edits can be tracked through the “Track Changes” option.
Comment 2: Discussion, page 9: please discuss the wide range of follow up, recurrence can be related to the assessment time after surgery.
Response: Thank you for your comment. We have added a sentence in the Discussion section for this part as follows: [Noteworthy, the follow-up duration of equinus patients varied remarkedly between individual studies ranging from 6 months to as high as 168 months after the interventional correction of equinus. Also, based on the qualitative assessment of retrieved data from individual studies, we noted a tendency for higher recurrence rates in studies with longer follow-up durations. This further complicates the issue of the actual recurrence rate after equinus correction in CP patients. Therefore, we recommend future studies to report the recurrence rate on both the short- and long-term.]
Comment 3: Furthermore we suggest to proceed on the possible analysis of subgroups: as recurrence is more common during growth, why didn’t you consider to divide groups according age and follow-up lenght?
Response: Thank you for this comment, that is an excellent question. First of all, before we extracted the data, we had that in mind, however, after retrieving the data and looking at it, it was not possible to carry subgroup analyses based on these 2 variables.
Regarding age, a subgroup analysis was not applicable because the mean age of patients wasn’t reported in 10 studies, and even in the remaining studies where age was reported, some studies reported the mean age without the standard deviation, and some studies reported the range without reporting the median and interquartile range. So, it was not possible to pool these data and perform a subgroup analysis on this variable.
As for the follow-up, it was also unfeasible to do a subgroup analysis on this factor because we couldn’t determine a reliable cut-off value to difference the reported data into (short-term or long-term).
Minor points
Comment 1: Line 34-35: “ Equinus deformity does not occur particularly due to contracture of the gastrocsoleus muscle” is it an error or do you mean that there is another more common cause of it?
Response: Thank you for highlighting this mistake. It was corrected as follows: [Equinus deformity occurs particularly due to contracture..]
Comment 2: Table 1 pag 5: you should describe the Intervention of Matsuo 2011 and Joo 2011
Response: Thank you for your comment. We have added the description of the surgeries performed for each of these studies in the revised version of the Table.
Comment 3: Table 1 pag 6: as multilevel surgery is not specific for equinus deformity, you should specify which is the specific intervention considered by Dreher 2012 for the equinus. The same is for Katz 2000.
Response: Thank you for your comment. We have edited the Table according to your suggestions.
Reviewer 3 Report
Thank you for opportunity to review this meta-analysis. I provide some comments.
The introduction needs to be supplemented. For example, there is statement of ‘Recurrence following equinus correction is reported in numerous studies.’ What kinds of studies were representative and what were the main research results should be revealed. In addition, it is necessary to mention how it relates to the purpose of this study.
Research questions or hypothesis should be stated.
Please, move PRISMA Flow Diagram for selection of included studies to method section.
Figures need corrections. Some studies are named as author and other studies are named as author/year/country. It is suggested to unify the authors and provide reference numbers.
In the discussion, please describe the implications of the results, not repeated statements of the results.
Please suggest what your findings suggest and how they should be reflected in future research.
Author Response
Comment 1: The introduction needs to be supplemented. For example, there is statement of ‘Recurrence following equinus correction is reported in numerous studies.’ What kinds of studies were representative and what were the main research results should be revealed. In addition, it is necessary to mention how it relates to the purpose of this study.
Response: Thank you so much for your comment. We have edited this part of the introduction to show more data as follows: [Recurrence following equinus correction is reported in numerous studies with rates differing widely from a study to another (ranging from 4.7 to 28.2%). There is a wide discrepancy in the reported rates of recurrence with no consistency in the rates reported in individual studies. In the same context,…].
Comment 2: Research questions or hypothesis should be stated.
Response: Thanks a lot for your comment. We added a new sentence highlighting our hypothesis at the end of the “Introduction section” as follows: [We hypothesize that the recurrence rate of equinus following its correction will be considerably high, both in surgically and non-surgically treated CP patients.]
Comment 3: Please, move PRISMA Flow Diagram for selection of included studies to method section.
Response: Thank you for your comment. We moved the PRISMA diagram to the Methods section.
Comment 4: Figures need corrections. Some studies are named as author and other studies are named as author/year/country. It is suggested to unify the authors and provide reference numbers.
Response: Thank you for your comment. All figures have been edited to provide a unified structure (last name of the first author/year of publication/country).
Comment 5: In the discussion, please describe the implications of the results, not repeated statements of the results.
Response: Thank you for your comment. We have actually responded to a very similar comment raised by Reviewer 1 (comment 2). In summary, before the limitation section, we added a new sentence discussing how our findings correlate with the current literature and how to interpret and implicate our findings in clinical practice, as follows: [Our study is the first to examine the rate of equinus recurrence in different interventions for the correction of equinus in patients with cerebral palsy. We highlight that many factors could play a role in the rate/risk of equinus recurrence following its correction, including clinicodemographic characteristics, such as laterality and the type of intervention.]
Comment 6: Please suggest what your findings suggest and how they should be reflected in future research.
Response: Thank you for your comment. We have discussed this part in detail in the previous comment and within the limitations section where we gave our recommendations and suggestions for future research to further investigate this area.